# Uncertainty Quantification of Soil Organic Carbon Estimation from Remote Sensing Data with Conformal Prediction

Nafiseh Kakhani [1,2,*], Setareh Alamdar [3], Ndiye Michael Kebonye [1,4], Meisam Amani [5] and Thomas Scholten [1,2,4]

1   Department of Geosciences, Soil Science and Geomorphology, University of Tübingen,
    72070 Tübingen, Germany
2   CRC 1070 RessourceCultures, University of Tübingen, 72070 Tübingen, Germany
3   School of Environmental Sciences, University of Guelph, 50 Stone Rd East, Guelph, ON N1G 2W1, Canada
4   DFG Cluster of Excellence "Machine Learning", University of Tübingen, 72070 Tübingen, Germany
5   WSP Environment and Infrastructure Canada Limited, Ottawa, ON K2E 7L5, Canada
*   Correspondence: nafiseh.kakhani@uni-tuebingen.de

**Abstract:** Soil organic carbon (SOC) contents and stocks provide valuable insights into soil health, nutrient cycling, greenhouse gas emissions, and overall ecosystem productivity. Given this, remote sensing data coupled with advanced machine learning (ML) techniques have eased SOC level estimation while revealing its patterns across different ecosystems. However, despite these advances, the intricacies of training reliable and yet certain SOC models for specific end-users remain a great challenge. To address this, we need robust SOC uncertainty quantification techniques. Here, we introduce a methodology that leverages conformal prediction to address the uncertainty in estimating SOC contents while using remote sensing data. Conformal prediction generates statistically reliable uncertainty intervals for predictions made by ML models. Our analysis, performed on the LUCAS dataset in Europe and incorporating a suite of relevant environmental covariates, underscores the efficacy of integrating conformal prediction with another ML model, specifically random forest. In addition, we conducted a comparative assessment of our results against prevalent uncertainty quantification methods for SOC prediction, employing different evaluation metrics to assess both model uncertainty and accuracy. Our methodology showcases the utility of the generated prediction sets as informative indicators of uncertainty. These sets accurately identify samples that pose prediction challenges, providing valuable insights for end-users seeking reliable predictions in the complexities of SOC estimation.

**Keywords:** uncertainty quantification; conformal prediction; soil organic carbon; digital soil mapping; remote sensing

## 1. Introduction

The soil represents the most extensive reservoir of terrestrial organic carbon within the biosphere, containing a greater amount of carbon than that found in both plants and the atmosphere [1]. Soils also play a crucial role in the sequestration of atmospheric carbon dioxide, $CO_2$, as well as in the emission of trace gases that contribute to the greenhouse effect [2]. Soil organic carbon (SOC) is one of the most important soil properties that serves as a crucial indicator of soil health and plays a significant role in supporting essential ecosystem services, such as nutrient cycling and food production [3]. Various environmental factors, including droughts, fires, increased level of heavy metals, alterations in land use patterns, and the anticipated consequences of global warming, can significantly influence organic carbon stocks, exerting a profound impact on the Earth's ecosystem [4–6]. In light of these potential consequences, the importance of monitoring SOC levels is a priority. This monitoring process stands out as one of the most effective methods for assessing ecosystem health, protecting biodiversity, and preserving natural habitats [7].

Monitoring and assessing SOC involves the utilization of remote sensing methods, which employ satellite, airborne, and ground-based sensors. These techniques include various approaches. For instance, in [8], lidar technology is utilized to provide high-resolution elevation data, characterizing vegetation structure and biomass, thereby offering insights into SOC levels. Another method involves infrared and thermal imaging, where sensors capture heat emissions and surface temperatures. This technique, as discussed in [9], correlates changes with organic carbon content, providing a quantitative assessment. Additionally, proximal sensing utilizes ground-based sensors, which are beneficial for close-range measurements and validating remote sensing results (e.g., [10]). This comprehensive suite of remote sensing techniques facilitates a thorough understanding of SOC across diverse spatial scales and environmental conditions. In recent years, the soil science community has seen the emergence of digital soil mapping (DSM) techniques, which aid in comprehending the spatial distribution of soil properties, particularly SOC [11]. DSM generates soil maps by employing statistical inferences derived from a prediction model. This model utilizes an extensive set of available environmental covariates that can be provided via remote sensing as predictors and is trained using soil sample data [12,13].

Although DSM is a widely employed and popular technique, the predictions it produces are not free from errors [14]. These errors originate from multiple sources, with a primary contributor being the inherent limitations of the input data [15,16]. Notably, environmental features often prove insufficient in elucidating the entirety of soil variations [17], lacking the desired level of informativeness. The applied model itself introduces another source of error, as it has the potential to propagate errors from inputs to final outputs [18]. Moreover, the intricate relationships between environmental features and soil properties may not always be comprehensively captured by the model, leading to prediction errors. Additionally, imprecise measurement methods and the utilization of small training sample sizes can further contribute to the presence of errors in predictions [19].

Various approaches have been employed to quantify uncertainty in the field of DSM. Geostatistical modeling techniques, such as kriging models [20], kriging with external drift [21], and the use of empirical models [22], offer a means to evaluate uncertainty levels by facilitating the computation of spatial representations for prediction intervals. It is important to note, however, that these uncertainty assessments are inherently linked to the specific modeling framework employed and may not seamlessly align with the increasingly prevalent use of ML techniques in DSM applications [23]. The bootstrapping method offers an alternative approach to uncertainty estimation, as exemplified in [14,24]. This technique entails training a model with a randomly selected subset of the complete training samples, followed by the generation of multiple predictions. These predictions play a crucial role in deriving uncertainty estimates, achieved by computing the average mean squared error (MSE) across all predictions and integrating it with the prediction variance calculated during the bootstrapping process for each prediction. However, it is essential to recognize that the practical applicability of this method may be constrained, particularly when dealing with large datasets, due to computational limitations. This limitation arises from the necessity to predict the entire extent of each map realization to estimate prediction variance [25].

Lately, there has been a growing interest in the field of DSM in the use of ML algorithms that have the capability to predict conditional quantiles, as evidenced by the works of [26–28]. Two examples of such techniques include the quantile regression forest introduced in [29] and the quantile regression neural network presented by [30]. These methods represent probabilistic adaptations of a random forest and an artificial neural network, respectively. Alternative uncertainty estimation approaches facing computational challenges include Monte Carlo simulation [31] and the Gaussian process [32]. It is noteworthy that the Gaussian method primarily quantifies uncertainty associated with input data, whereas the Monte Carlo method is specifically tailored for assessing uncertainty in parameters [33]; hence, each method exhibits distinct limitations. In addition to the

aforementioned limitations, none of the previously discussed methods provide statistical guarantees that the final output is highly probable to be included in the prediction sets.

Irrespective of the origins of errors, establishing confidence in DSM products, typically represented as maps, is crucial for end-users. However, this trust is hindered by considerable challenges arising from inherent errors within these products. The deficit in trust is further compounded by a lack of comprehensive understanding of the underlying models and the absence of performance guarantees, as noted in recent studies [34,35]. Consequently, end-users express concerns not only about the overall accuracy of the map but also seek precise information regarding the accuracy of predictions at specific locations within the mapped study area. The diverse and complex nature of these challenges underscores the imperative for enhanced transparency and interpretability in DSM models [36], generalizations to new observations [37], robustness to out-of-distribution observations [38], and the quantification of uncertainty [39].

In addressing one aspect of this multi-dimensional issue, our focus is directed toward the domain of uncertainty quantification and management. Conformal prediction, recognized as a distribution-free method for uncertainty quantification, addresses this concern by specifically endeavoring to offer assurances regarding sample coverage. Precisely, given a specified tolerance error or confidence level $a \in [0,1]$, the conformal approach allows for a statistical guarantee that the true value lies within the prediction intervals. The efficacy of the conformal approach has been demonstrated across various fields. For instance, in [25], it was applied to quantify uncertainty in land use land cover classification at the pixel level. In a recent study, inductive conformal prediction has been introduced for an image-based harvest readiness classification that can select a pre-defined level of predictive confidence in the model [40]. In another study, authors utilized conformal prediction for crop and weed classification through group-conditional conformal statistics, employing quantile regression on group membership indicators [41]. Additionally, [42] showcased the application of the ensemble version of conformal prediction for time series forecasting.

This study, for the first time, presents an introduction to conformal prediction [43,44] as a reliable and effective approach for assessing uncertainty in the prediction of SOC within a regression framework. In the context of regression, this framework enables a predictive model to generate prediction intervals or sets for a given observation $X$ rather than singular predictions, ensuring that the true value $Y$ falls within the prediction interval with a high probability [45]. This method offers notable advantages, primarily attributable to three key factors. Firstly, it demonstrates computational efficiency and scalability, particularly in large-scale studies. Additionally, the approach exhibits adaptability to a broad spectrum of regression algorithms, encompassing various machine learning (ML) techniques. Moreover, the implementation of conformal prediction requires minimal preassumptions to construct predictive sets, facilitating effective empirical coverage. Conformal prediction, as far as our understanding goes, is a novel addition to the field of DSM. Consequently, the principal contribution of this paper lies in presenting conformal prediction to the DSM community, addressing uncertainties in ML decision-making procedures, and providing empirically demonstrable coverage guarantees.

The article is structured as follows. In Section 2, the mathematical framework defining conformal prediction and its application in calculating uncertainty within ML frameworks is established. Section 3 introduces the data, including ground reference samples and the required input features for establishing the predictive model. Section 3 also outlines the experimental setup, the implemented regression methods, and common techniques for uncertainty estimation in DSM. It also introduces evaluation metrics for both uncertainty and accuracy. Section 4 delves into the experimental results and compares different implemented methods. Finally, Section 5 provides a comprehensive discussion and interpretation of derived uncertainty, spatial distribution of samples with high and low uncertainty, and empirical coverage of samples within different land cover classes. The concluding remarks of the study are presented in Section 6.

## 2. Mathematical Background

Conformal prediction is a probabilistic framework that creates valid prediction intervals using a similarity metric called 'conformity' [43]. It does not rely on assumptions about distribution, except for the requirement that observations must be exchangeable, meaning that the order of observations does not affect the information they provide.

Consider $\{(X_i, Y_i)\}_{i=1}^n$ to be a pair of predictors, $X_i \in \mathbb{R}^d$, and response variables, $Y_i \in \mathbb{R}$. We split it into two subsets: the training set, $\mathcal{S}_1$, and a validation set, $\mathcal{S}_2$. First, we build a regression model using $\mathcal{S}_1$, and then the conformity score, a measure of the prediction errors (residuals) derived from $\mathcal{S}_2$, which is employed to assess the level of uncertainty in forthcoming predictions [42].

When presented with a new observation $X_{n+1} = x$, we ensure that the conditional prediction intervals for the target variable $Y_{n+1}$, denoted as $\hat{C}_a(x)$ and characterized by a miscoverage rate $a$, satisfy the following condition:

$$\mathbb{P}\{Y_{n+1} \in \hat{C}_a(X_{n+1} = x)\} \geq 1 - a. \tag{1}$$

Conformal prediction offers conditional prediction intervals, $\hat{C}_a(x)$, in the following manner:

$$\hat{C}_a(x) = [\hat{\mu}(x) - q_{1-a}(\mathcal{R}, \mathcal{S}_2), \hat{\mu}(x) + q_{1-a}(\mathcal{R}, \mathcal{S}_2)], \tag{2}$$

where $\hat{\mu}(x)$ represents the prediction generated by the underlying regression model or base model. $\mathcal{R}$ denotes the collection of residuals $R_i$, which are calculated based on the predictions of the samples indexed by $i \in \mathcal{S}_2$. The conformity score $q_{1-a}(\mathcal{R}, \mathcal{S}_2)$ corresponds to the $(1-a)$-th quantile of $\mathcal{R}$ [46]. The residuals used for computing the conformity score are typically determined using the $L_1$ norm, although alternative distance metrics may also be employed. Equation (2) reveals that conformal prediction, in this form, was originally designed with the assumption of homoscedastic data in mind. Homoscedasticity refers to the property of having constant variance across the range of predictor variables in a statistical model. In the context of conformal prediction, this assumption implies that the variability in the data remains uniform, regardless of the values of the predictors. This is because the prediction interval is constructed as a conditional mean estimate of the response variable, surrounded by a fixed-width band [47].

A quantile regression can be used as the base model to produce predictions. In this regard, quantile regression seeks to estimate the conditional quantile function (CQF) of $Y$ given $X$ for a specified value of $(0 < a < 1)$. It is formally defined as follows:

$$q_a(x) = \inf\{y \in \mathbb{R} : F_Y(y \mid X = x) \geq a\}, \tag{3}$$

where $F_Y(y)$ is the conditional distribution of $Y$. Prediction intervals can be derived directly from two empirical CQFs calculated from the training dataset. The confidence level for these prediction intervals, denoted as $(1-a)$, corresponds to the difference between these two quantile levels. Consequently, the estimated conditional prediction interval for quantile regression is defined as follows:

$$\hat{C}_a(x) = \left[\hat{q}_{a_{lo}}(x), \hat{q}_{a_{hi}}(x)\right], \tag{4}$$

where $\hat{q}_{a_{lo}}(x)$ and $\hat{q}_{a_{hi}}(x)$ are the empirical CQFs computed for $a_{lo} = a/2$ and $a_{hi} = 1 - a/2$. Unlike Equation (2), the width of the prediction interval in Equation (4) depends on each specific data point $x$ and can vary significantly from point to point. Therefore, quantile regression yields intervals that adapt to heteroscedasticity in the data. However, when the ideal interval $C_a(x)$ is replaced by the finite sample estimate $\hat{C}_a(x)$ in Equation (4), the actual coverage of the prediction interval is not guaranteed to match the designed confidence level $(1-a)$ [46].

The estimation of $\hat{q}_{a_{lo}}(x)$ and $\hat{q}_{a_{hi}}(x)$ could be seen as an optimization problem that minimizes a loss function and can be defined as:

$$\rho_{a,i} = \begin{cases} (1-a)(\hat{q}_a(x_i) - y_i), & \hat{q}_a(x_i) - y_i \geq 0; \\ a(y_i - \hat{q}_a(x_i)), & otherwise, \end{cases} \tag{5}$$

in which $y_i$ denotes the $i$-th sample response and $q_a(x_i)$ is the $a$-th quantile estimated from the quantile regression model. The simplicity and versatility of this approach make it suitable for a broad range of applications. Similar to traditional regression analysis, different ML models can be applied to create and train the loss function (e.g., [48,49]).

Now we can take advantage of both conformal prediction and quantile regression for prediction. The properties of quantile regression allow the method to adapt to the local variability in the data, and the use of conformal prediction guarantees valid marginal coverage. After training the quantile regression algorithm, we can define the conformity scores $E_i$, to be the difference between $y$ and its nearest quantile. In other words, the resulting prediction intervals are conformalized using the conformity scores:

$$E_i = \max\{\hat{q}_{a_{lo}}(x_i) - y_i, y_i - \hat{q}_{a_{hi}}(x_i)\}, i \in \mathcal{S}_2 \tag{6}$$

The scores are evaluated on the validation set, $\mathcal{S}_2$, and quantify the error made by the prediction interval of the regression algorithm, as specified by the lower and upper bounds $\hat{q}_{a_{lo}}(x_i)$ and $\hat{q}_{a_{hi}}(x_i)$ in Equation (4). Given a new input $X_{n+1}$, the prediction intervals are calculated via:

$$\hat{C}_a(x) = [\hat{q}_{a_{lo}}(x) - Q_{1-a}(\mathcal{E}, \mathcal{S}_2),$$
$$\hat{q}_{a_{hi}}(x) + Q_{1-a}(\mathcal{E}, \mathcal{S}_2)], \tag{7}$$

where $\mathcal{E} = \{E_i\}_{i \in \mathcal{S}_2}$. The value of $Q_{1-a}(\mathcal{E}, \mathcal{S}_2)$, utilized to transform the prediction intervals created by the chosen quantile regression technique, remains constant for all new data points $x$, just like how $q_{1-a}(\mathcal{R}, \mathcal{S}_2)$ behaves in the context of conformal prediction in Equation (2) [50].

In more straightforward terms, we can express that the sets $\hat{C}_a(x)$ either expand or contract the gap between the quantiles by $Q1 - a(\mathcal{E}, \mathcal{S}_2)$ to attain the desired coverage in Equation (1), as illustrated in Figure 1.

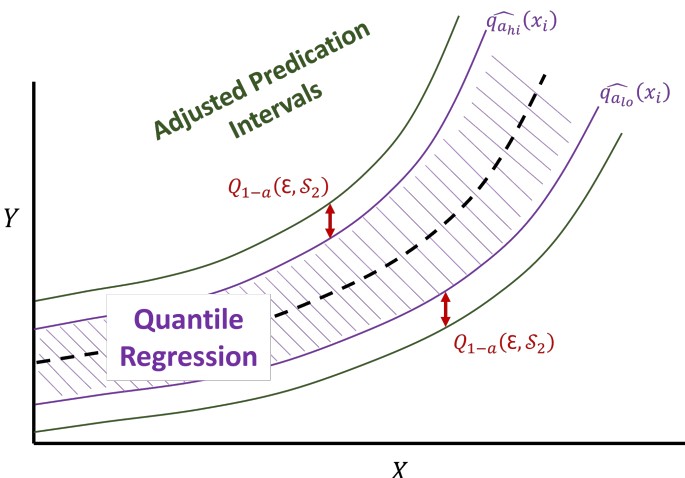

**Figure 1.** Conformalized quantile regression in which the quantiles are adjusted by the constant $Q_{1-a}(\mathcal{E}, \mathcal{S}_2)$ calculated via the validation set.

## 3. Materials and Methodology

### 3.1. Data Description and Preprocessing

To construct a predictive model, it is essential to obtain ground reference samples containing information on SOC content. The details of this ground reference information can

be found in Section 3.1.1. Alongside this dataset, input features, which are comprehensively elucidated in Section 3.1.2, are required.

### 3.1.1. Ground Reference Samples

Established in 2001, the Land Use/Cover Area frame Survey (LUCAS) program initially employed a visual assessment area frame survey tailored for agricultural policies. Evolving in 2006 to adopt a systematic grid structure with 22 km grid cells across the EU territory, the program took a significant step with the incorporation of the specialized "LUCAS-Topsoil" component. Developed in collaboration with Eurostat, DG ENV (the Directorate-General for Environment), and JRC (the Joint Research Centre), this component focuses on evaluating topsoil within the 0 to 20 cm depth range. The LUCAS topsoil dataset, comprising comprehensive information on soil properties, such as texture, organic carbon content, pH levels, and nutrient concentrations, stands as a valuable resource for researchers, policymakers, and land managers. It also plays a crucial role in advancing agricultural practices, environmental monitoring, and sustainable land management strategies at a continental scale. In this study, the 2015 LUCAS dataset, which provides data on SOC for all European Union member countries, was employed. These countries could be categorized into eight different climate zones: boreal to sub-boreal, Atlantic, sub-oceanic, sub-continental northern and southern, temperate mountainous, and Mediterranean (semi-arid and temperate to sub-oceanic). Table 1 presents the statistical details of the data for reference. The distribution of SOC values is also depicted in Figure 2.

The LUCAS dataset is a rich collection of accessible soil samples. This abundance of information facilitates a comprehensive exploration of soil characteristics, contributing to the resilience of our investigation. Moreover, it can assist in modeling soil properties on a continental scale, thereby augmenting the precision and clarity of our analysis. Table 1 presents the statistical details of the data for reference. The distribution of SOC values is also depicted in Figure 2.

**Table 1.** Statistical summaries of the LUCAS soil samples used in this study.

|  | **Mean** | **s.d.** | **Min.** | **Q1** | **Median** | **Q3** | **Max.** |
|---|---|---|---|---|---|---|---|
| SOC (g/kg) | 43.27 | 76.70 | 0.1 | 12.5 | 20.4 | 38.6 | 560.2 |

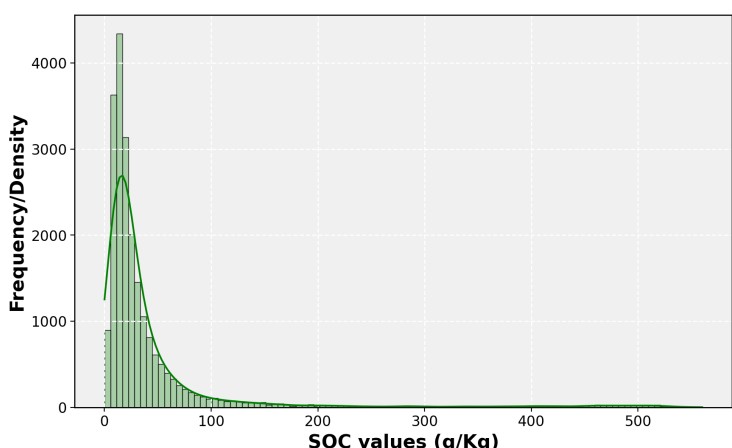

**Figure 2.** Histogram and kernel density estimation plot depict the distribution of SOC (g/kg) values.

### 3.1.2. Input Features
Climate Data

To investigate the environmental determinants that either facilitate or hinder climate regulation, we utilized a comprehensive array of parameters sourced from TerraClimate [51]. Monthly time series were generated based on gridded meteorological data through spa-

tiotemporal interpolation of the WorldClim datasets [52]. This study incorporated variables, such as temperature, evapotranspiration, precipitation, soil moisture, surface radiation, and vapor pressure, which are relevant environmental factors known for their impact on SOC dynamics [52,53] and their significance in the processes contributing to the ecosystem's climate regulation capacities [54,55]. For data acquisition, we obtained a subset covering the temporal interval of 2010 to 2015. In cases where the dataset exhibited missing values (NaN), K-nearest neighbors (KNN) interpolation was applied to ensure temporal completeness of records. For each specific climatic feature, we computed both the mean and the difference between maximum and minimum values over the 5-year interval.

Landsat-8 Bands

Our research utilized Landsat-8 surface reflectance data from the year 2015, which was selected based on its proximity to our sampling period. Our analysis was designed to minimize the impact of vegetation by selecting images captured during winter season (i.e., November/December and January/February), which are less likely to be affected by dense vegetation. If we were unable to find a suitable image from the winter months, the image with the lowest NDVI value from the other months of the sampling year was selected. In addition, we implemented a filtering process to eliminate images with substantial cloud cover, shadow, or snow, making sure they accounted for less than 10 percent of the total area of the image. As a result of this meticulous data collection approach, we were able to acquire soil-specific images for analysis, minimizing potential confounding factors and enhancing the relevance of the data collected. However, in the case of sampling locations within forests or permanent grasslands, access to soil reflectance remains a plausible challenge. To gather the required data for each sample point, we downloaded Landsat image patches centered around each sample location, using a window size of 3 by 3 pixels.

Vegetation and Mineral Indices

As part of our research, a collection of indices that cover a range of soil and vegetation characteristics was developed. Several indices, including clay minerals [56], ferrous minerals [57], carbonate index, rock outcrop index, and Normalized Different Vegetation Index (NDVI), provided information about soil and vegetation attributes within the study region, particularly with respect to SOC [58,59].

Topography

Digital elevation data sourced from the Shuttle Radar Topography Mission (SRTM) were also utilized in this study. These data, part of a global research initiative, aim to create digital elevation models covering extensive geographic areas. The SRTM-V3 product (SRTM Plus) provided by NASA JPL offers a resolution of 1 arc-second, approximately equal to 30 m. Alongside elevation data, we incorporated information on slope, valley bottom flatness (VBF)—a ratio indicating the flatness of the valley bottom concerning a reference surface—and topographic wetness index (TWI)—representing the degree of landscape wetness based on its topography. These variables play a pivotal role in shaping soil distribution across the landscape by influencing processes like overland flow and erosion, ultimately impacting the behavior, as observed in [59,60]. By concentrating on these specific topographic factors, our objective is to comprehensively understand the intricate relationship between terrain characteristics and SOC dynamics within the designated study area. For a comprehensive understanding of the utilized features, including both summary and detailed information, refer to Tables 2 and 3. A detailed explanation of the preprocessing steps can also be found in [61].

**Table 2.** The number of input features within each category and their corresponding temporal and spatial resolutions.

| Category | Number of Features | Spatial Resolution | Temporal Resolution |
|---|---|---|---|
| Climate Data | 12 | ∼4 km | One month |
| Landsat-8 Bands | 7 | 30 m | 16 day |
| Vegetation and Mineral Indices | 5 | 30 m | 16 day |
| Topography | 4 | 30 m | One time mission |

**Table 3.** The specification of input features, including remote sensing images and spectral indices, topographical characteristics, and climate features, along with their corresponding units of measurement.

| No | Feature | Description | Unit |
|---|---|---|---|
| 1 | L8B1 | Ultra Blue | $W/m^2 \cdot sr \cdot \mu m$ |
| 2 | L8B2 | Blue | $W/m^2 \cdot sr \cdot \mu m$ |
| 3 | L8B3 | Green | $W/m^2 \cdot sr \cdot \mu m$ |
| 4 | L8B4 | Red | $W/m^2 \cdot sr \cdot \mu m$ |
| 5 | L8B5 | NIR | $W/m^2 \cdot sr \cdot \mu m$ |
| 6 | L8B6 | SWIR1 | $W/m^2 \cdot sr \cdot \mu m$ |
| 7 | L8B7 | SWIR2 | $W/m^2 \cdot sr \cdot \mu m$ |
| 8 | Clay Minerals | $(SWIR1 - SWIR2)/(SWIR1 + SWIR2)$ | Unitless |
| 9 | Ferrous Minerals | $(NIR - SWIR1)/(NIR + SWIR1)$ | Unitless |
| 10 | Carbonate Index | $(Red - Green)/(Red + Green)$ | Unitless |
| 11 | Rock Outcrop Index | $(SWIR1 - Green)/(SWIR1 + Green)$ | Unitless |
| 12 | NDVI | $(NIR - Red)/(NIR + Red)$ | Unitless |
| 13 | Elevation | Elevation | m |
| 14 | Slope | Slope | Percent |
| 15 | VBF | Vally bottom flatness | Unitless |
| 16 | TWI | Topography wetness index | Unitless |
| 17 | Actual evapotranspiration | Actual evapotranspiration | mm |
| 18 | pdsi | Palmer Drought Severity Index | Unitless |
| 19 | Climate water deficit | Climate water deficit | mm |
| 20 | Reference evapotranspiration | Reference evapotranspiration | mm |
| 21 | Precipitation accumulation | Precipitation accumulation | mm |
| 22 | Soil moisture | Soil moisture | mm |
| 23 | Surface radiation | Downward surface shortwave radiation | $W/m^2$ |
| 24 | Minimum temperature | Minimum temperature | °C |
| 25 | Maximum temperature | Maximum temperature | °C |
| 26 | Vapor pressure deficit | Vapor pressure deficit | kPa |
| 27 | Vapor pressure | Vapor pressure | kPa |
| 28 | Wind speed | Wind speed at 10 m | m/s |

*3.2. Experiments*

The objective of our experiments is to assess the effectiveness of conformal prediction for SOC estimation and to compare it with commonly used methods for uncertainty quantification in DSM. We conceptualize our quantile regression model as a quantile random forest and tailor the prediction intervals according to the previously outlined reasoning in Section 2. To this end, all ground reference samples were initially divided into training and test groups for five different seeds. Subsequently, we employed the training samples for analysis, reserving the test samples exclusively for the final evaluation. Furthermore, the training data were partitioned into training and validation groups over ten iterations. During these iterations, all training steps were executed for all implemented models, with the remaining validation samples serving different purposes based on the selected methodology. For instance, in the case of the quantile versions of random forest and gradient boosting (QRF and QGB), which required hyperparameter selection, we determined the optimal parameters based on the validation samples. For QGB, these

parameters include learning rate, number of estimators, maximum depth, and minimum sample split. For QRF, the hyperparameters are the number of estimators and minimum sample leaf.

Additionally, the conformal prediction involved an extra step of calculating conformity scores using the validation set. The miscoverage rate *a* for this step was set to 0.1. In the case of the quantile neural network (QNN), we utilized the validation set to prevent overfitting of the network and to stop the training loop. For quantile linear regression (QLR), the set was used to identify the best model performance across different iterations. As for bootstrapping random forest (BRF), we performed resampling using the training samples ten times without any further validation steps. A comprehensive description of the implemented method is presented in Section 3.2.1. The experimental setup is visually represented in Figure 3.

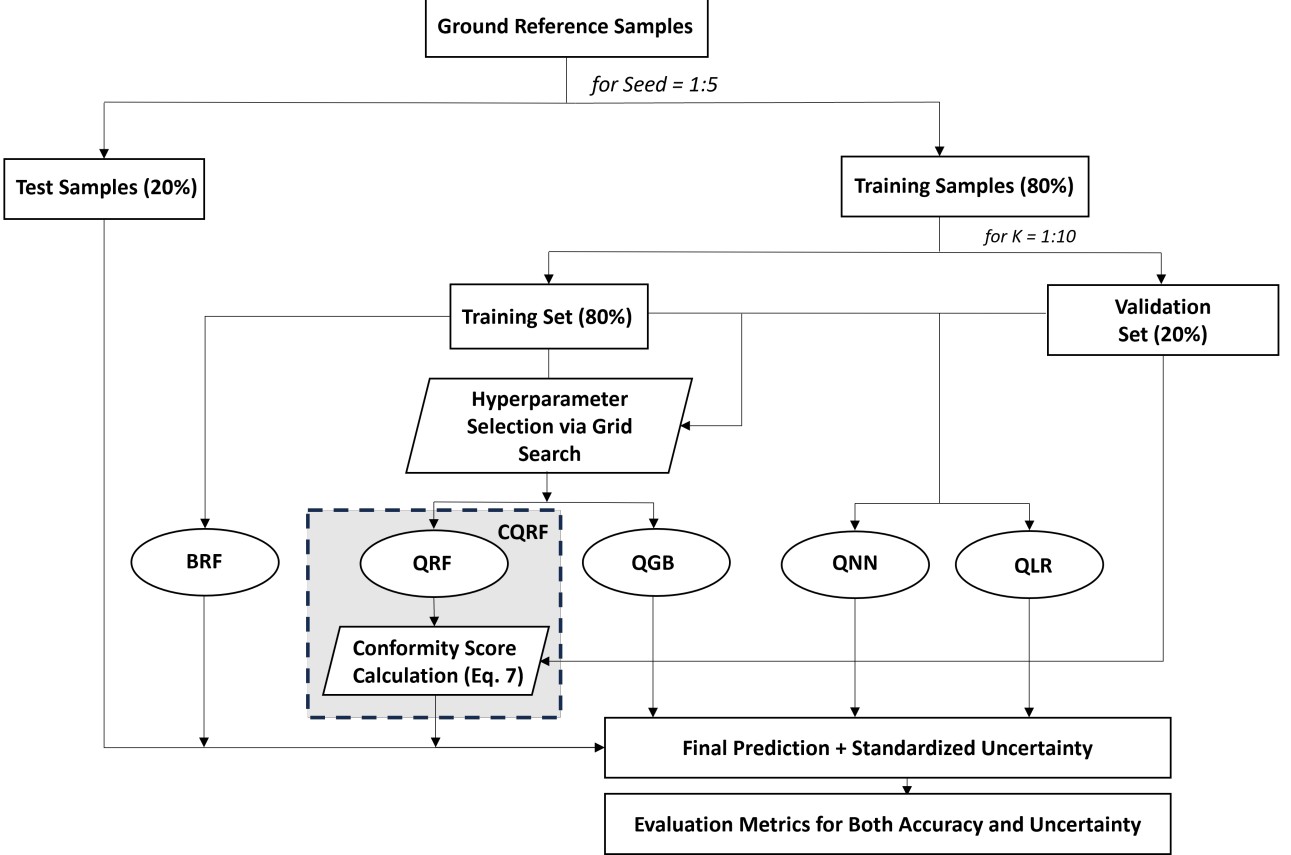

**Figure 3.** The experimental setup outlines the procedural aspects of data partitioning, model implementation, and the evaluation metrics.

### 3.2.1. Implemented Methods

We applied five distinct methods, which will be elaborated upon in the subsequent sections. All experiments were conducted using Python, with the utilization of packages such as `Sci-Kit Learn`, `Sci-Kit Garden`, and `Statsmodels`.

### Bootstrapping Random Forest

An RF regression approach coupled with bootstrapping was applied to comprehensively assess and quantify uncertainty in our predictive modeling framework. RF Regression, a powerful ensemble learning technique, was chosen for its ability to handle complex relationships within the multi-modal data by aggregating the predictions of multiple decision trees [62]. To enhance the robustness of our predictions and provide a thorough characterization of the uncertainty associated with our model, we implemented

bootstrapping—a resampling technique that involves generating multiple bootstrap samples from the original dataset. By constructing multiple models on these resampled datasets and subsequently aggregating their predictions, we were able to derive a distribution of outcomes, allowing for the estimation of prediction intervals and enhancing our understanding of the inherent variability in the model predictions.

### Conformalized Quantile Random Forest

As mentioned in the previous section, the RF constitutes a prevalent regression technique in the DSM field. To implement this methodology following the guidelines outlined in Section 2, we instantiated the quantile random forest as the conditional quantile function, as detailed in Equation (3). The training process involves utilizing training samples. Following this, we applied conformal prediction to the validation set (Equation (7)), enabling the adjustment of prediction intervals for each specific outcome. These intervals act as indicators of the inherent uncertainty associated with the predictive outcomes.

### Quantile Neural Networks

QNNs are a class of neural networks designed for estimating conditional quantiles of a target distribution, here SOC, and they often utilize the *pinball loss* function as an integral component of their training process. *Pinball loss* is a quantile-specific loss function that penalizes the differences between predicted and observed quantiles. By incorporating the *pinball loss* function, QNNs are tailored to optimize the parameters of the network to produce accurate quantile estimates across a range of quantiles, not just a single-point estimate. This approach enables QNNs to capture the spectrum of uncertainty associated with the data.

### Quantile Gradient Boosting

QGB extends traditional gradient boosting methods to address quantile regression tasks. QGB achieves this by iteratively fitting a series of weak learners, typically regression trees, to minimize a specialized loss function designed for quantiles. The loss function measures the differences between predicted and observed quantiles and is optimized during each boosting iteration. This approach allows QGB to adapt to the specific characteristics of the data distribution.

### Quantile Linear Regression

QLR is a statistical method that extends linear regression by modeling not only the conditional mean but also various quantiles of the response variable. Unlike traditional regression, which focuses only on the mean value, quantile regression allows for a more comprehensive analysis of the distribution by estimating different quantiles, such as the median or other percentiles. In this study, we estimated the 5th and 95th quantiles.

### 3.2.2. Evaluation Metrics

While numerous metrics are accessible, determining a golden metric for uncertainty quantification remains an ongoing research question [63]. Our approach suggests value in examining multiple metrics simultaneously. Therefore, we have categorized our evaluation metrics into those associated with uncertainty in Section 3.2.2 and others related to the accuracy of prediction models in Section 3.2.2.

### Uncertainty

1. Negative Log-Likelihood (NLL): NLL measures the agreement between predicted ($y_{\text{pred}}$) and observed values ($y_{\text{obs}}$) under the assumption of a Gaussian distribution with a mean of zero and a standard deviation ($\sigma$). The NLL is defined as:

$$\text{NLL} = -\sum \ln\left( \frac{1}{\sigma\sqrt{2\pi}} e^{-\frac{(y_{\text{obs}} - y_{\text{pred}})^2}{2\sigma^2}} \right) \tag{8}$$

2.  Interval Score (IS): The scoring function $IS(\beta)$ is designed to evaluate the performance of a predictive model's interval predictions. It considers the average width of prediction intervals and introduces penalties for observations falling outside the predicted intervals. The parameter $\beta$ and $\delta$ play roles in determining the weighting and conditions for the penalties.

$$
\begin{aligned}
IS(\beta) = & \frac{1}{n} \sum_{i=1}^{n} (U_i - L_i) \\
& + \frac{2}{1-\beta} \cdot \frac{1}{n} \sum_{i=1}^{n} (L_i - y_i) \cdot \delta(y_i < L_i) \\
& + \frac{2}{1-\beta} \cdot \frac{1}{n} \sum_{i=1}^{n} (y_i - U_i) \cdot \delta(y_i > U_i).
\end{aligned}
\tag{9}
$$

The first term of Equation (9) represents the average width of prediction intervals for $n$ number of samples. The second term introduces a penalty for observations, $y_i$, that fall below the lower bound, $L_i$, of the prediction interval. The penalty is proportional to the difference between the lower bound and the actual observation. The factor, $2/(1-\beta)$, scales the penalty, and $\delta(\cdot)$ is the indicator function, which equals 1 if the condition inside is true and 0 otherwise. Similar to the second term, the third term penalizes observations that exceed the upper bound, $U_i$, of the prediction interval.

3.  Prediction Interval Coverage Probability (PICP): The PICP is a fundamental metric used to assess the reliability and calibration of prediction intervals. It quantifies the proportion of observed data points that fall within the model's prediction intervals. In simpler terms, it shows the coverage of samples. A well-calibrated model would ideally have a PICP close to the specified confidence level, indicating that a given percentage of prediction intervals should encompass the true values. The PICP is calculated as follows:

$$
\text{PICP} = \frac{1}{n} \sum_{i=1}^{n} c_i, \quad c_i = \begin{cases} 1, & y_i \in [L_i, U_i] \\ 0, & y_i \notin [L_i, U_i] \end{cases}
\tag{10}
$$

Like Equation (9), $U$ and $L$ are the upper and lower bounds of the prediction intervals and $n$ is the number of samples. A high PICP indicates that a significant portion of the observed data falls within the predicted intervals, reflecting well-calibrated and reliable predictions. Conversely, a low PICP suggests that the prediction intervals may be too narrow, indicating a potential lack of calibration in the model's uncertainty estimates.

4.  Prediction Interval Normalized Average Width (PINAW): The PINAW measures the normalized average width of the prediction intervals relative to the spread of the true values. It provides an indication of how well the width of the prediction intervals corresponds to the variability in the observed data. A lower PINAW indicates that the prediction intervals are narrower compared to the variability of the data.

$$
\text{PINAW} = \frac{1}{nD} \sum_{i=1}^{n} (U_i - L_i), \quad D = y_{\max} - y_{\min}
\tag{11}
$$

Accuracy Assessment

1.  Root Mean Square Error (RMSE): RMSE is widely used in statistics and data analysis for the accuracy assessment of a model. An accurate model can be assessed by calculating the square root of the mean of the squared differences, which quantifies the average magnitude of prediction errors.

$$
\text{RMSE} = \sqrt{\frac{1}{n} \sum (y_{\text{obs}} - y_{\text{pred}})^2}
\tag{12}
$$

2. Mean Absolute Error (MAE): MAE measures how well predictions or estimates match observed values. MAE calculates the difference between predicted or estimated values and observed values by taking the average of absolute differences instead of squared differences, as does RMSE.

$$\text{MAE} = \frac{1}{n} \sum |y_{\text{obs}} - y_{\text{pred}}| \tag{13}$$

3. Ratio of Performance to Interquartile distance (RPIQ): This metric represents the spread of the population and is calculated using the following equation [64]:

$$\text{RPIQ} = \frac{Q_3 - Q_1}{\text{RMSE}} \tag{14}$$

The values $Q_1$ and $Q_3$ represent the 25th and 75th percentiles of the observed samples, respectively, defining the interquartile distance.

**4. Results**

Table 4 presents the outcomes for various implemented methods. Given the use of five different seeds for test samples, the mean values for all metrics have been reported. The units for all metrics except PICP are consistent with the data and expressed in g/kg. The optimal results are highlighted in bold. The following insights can be derived.

**Table 4.** Evaluation metrics for both accuracy and uncertainty of the implemented models.

| Method | Uncertainty | | | | Accuracy | | | Final Score |
|---|---|---|---|---|---|---|---|---|
| | NLL | IS | PICP (%) | PINAW | MAE | RMSE | RPIQ | |
| BRF | 32.06 | 188.89 | 25 | 15.67 | **26.07** | **69.03** | **0.38** | 0.33 |
| CQRF | **4.93** | **143.75** | **90** | 111.91 | 42.20 | 80.79 | 0.33 | **0.31** |
| QNN | 5.89 | 243.99 | 89 | 165.58 | 72.05 | 111.73 | 0.27 | 0.58 |
| QGB | 5.06 | 155.75 | 89 | 124.30 | 48.15 | 82.39 | 0.32 | 0.35 |
| QLR | 550.87 | 419.54 | 2 | **2.24** | 41.145 | 86.59 | 0.30 | 0.64 |

All metrics are in g/kg except PICP (%) and RPIQ (unitless).

*4.1. Coverage and Prediction Interval Width: PICP and PINAW*

As outlined in Section 2, conformal prediction is characterized by its coverage guarantee, and CQRF exhibits the most average coverage across all seeds (PICP = 90%). Among other methods, both QGB and QNN also demonstrate good coverage, albeit at 89%, which is still less than CQRF. When considering PICP, the meaningfulness of the width of prediction intervals, PINAW, is also essential, as a method might return intervals from the minimum to the maximum of the dataset, achieving coverage for all samples but lacking meaningful prediction intervals. Therefore, there is a trade-off between PICP and PINAW.

By comparing CQRF, QGB, and QNN, which exhibit the best coverages, we observe that the smallest PINAW belonged to CQRF. For BRF and QLR, the PICP values are very low (25% and 2%, respectively), indicating that only a small percentage of the responses of the test samples fall within the prediction intervals returned by these methods. This deficiency arises because these methods lack consideration for sample coverage. Consequently, two significant findings are that (1) resampling may not be the optimal method for uncertainty quantification, and (2) while QLR is a straightforward and understandable method, it is evidently inadequate for uncertainty estimation. Given the low PICP for these methods, it is evident that the PINAW is also very low.

*4.2. Accuracy Metrics: RMSE, MAE, and RPIQ*

In the context of accuracy assessment metrics, which generally indicate the proximity of predicted values to observed values, the BRF method exhibits the lowest RMSE, MAE,



and RPIQ values. This is attributed to the method's exclusive focus on point estimation of outputs, albeit lacking the capability to generate validated uncertainty, as indicated by uncertainty metrics in Table 4. This underscores the notion that concentrating solely on the best method in terms of accuracy may compromise our ability in other aspects. The second most effective method is CQRF, demonstrating lower RMSE, MAE, and RPIQ values compared to those of QGB, QNN, and QLR. This suggests that CQRF not only provides better coverage but also delivers accurate predictions.

### 4.3. Scoring Rules: NLL and IS

Based on the results, QLR exhibits a remarkably high NLL value due to overfitting, a consequence of the non-normal distribution of the data (refer to Figure 2). BRF also presents a high NLL, attributed to the non-normal data distribution. However, BRF outperforms QLR because random forest demonstrates superior performance over linear regression when dealing with non-normal distributions. The NLL values for other methods, QNN and QGB, are comparable and reasonable, with CQRF exhibiting the best performance.

For the evaluation of prediction intervals using the scoring rule IS, optimal performance is observed for CQRF. This can be attributed to conformal prediction's attempt to calibrate the prediction interval while simultaneously ensuring coverage.

### 4.4. Summary of All: Final Score

In our analysis, comparing various metrics with distinct purposes posed a challenge for evaluating all implemented methods simultaneously. To streamline this comparison, we introduced a straightforward system to generate a final score. Initially, we transformed all metrics for all seeds to be negatively oriented, emphasizing smaller values as more desirable. For example, considering that PICP is positively oriented, we deducted its value from 100. Subsequently, we calculated the mean normalized values for all metrics within the range of 0 to 1. The method with the smallest value is considered the best performer.

While our study primarily focuses on uncertainty quantification, we chose to assign equal weights to all evaluation metrics in the normalization procedure. This decision is grounded in our belief that sacrificing accuracy solely for the sake of uncertainty quantification is not justifiable, ensuring a fair comparison. According to the final score, CQRF emerges as the top-performing method.

### 4.5. Variances of Metric Estimation

To precisely evaluate the accuracies of the implemented methods and to observe variations in metric estimation, we illustrated the results for different seeds in Figure 4. Notably, QLR exhibits the highest variance in NLL, primarily due to its large values for this metric. In contrast, PICP shows consistently small variances across all methods, suggesting that changes in the test sets do not significantly impact the study results.

A key observation comes to light when examining other metrics such as RMSE, MAE, RPIQ, IS, and PINAW. QNN displays the highest variances in the estimation of these metrics. This may be attributed to the utilization of the pinball loss function, designed to capture uncertainty. If the data distribution deviates from normal in a specific seed, it can lead to lower accuracy. In essence, the neural network proves highly sensitive to data imbalance, and this sensitivity manifests as elevated variances in metric estimation.

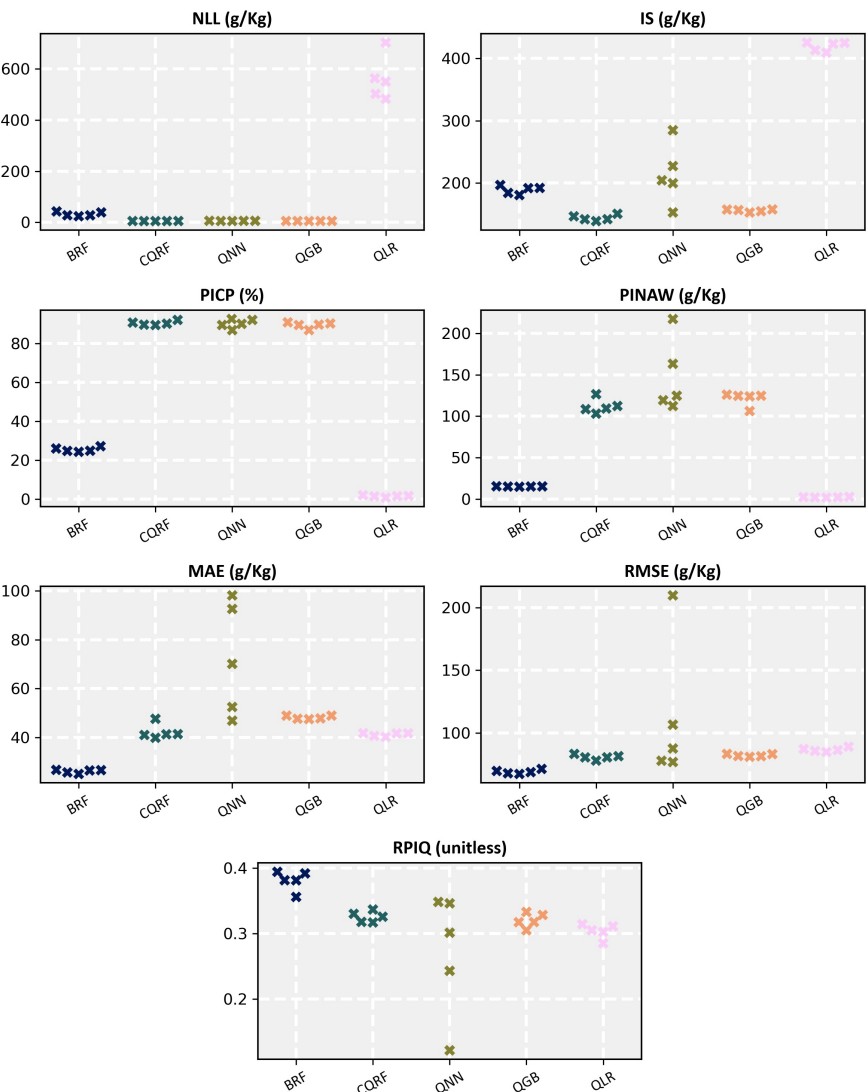

**Figure 4.** Performance metric variances across five seeds for the implemented models.

## 5. Discussion

### 5.1. Understanding Uncertainty in Environmental Contexts

To have a better understanding of the results, we decided to visualize the SOC values for all ground reference samples (Figure 5a) alongside standardized uncertainties. These uncertainties represent the standardized difference between lower and upper values for each sample, derived independently by each model (Figure 5b–f).

We aim to comprehend uncertainties related to our model based on well-known soil and environmental processes. Regarding the CQRF model, as depicted in Figure 5b, we noted elevated uncertainties linked to SOC estimates, particularly in regions characterized by very high altitudes (e.g., Austria, Italy), peatland deposits (e.g., Great Britain, Ireland), and boreal environments (e.g., Finland, Sweden, Estonia). These findings align with previous studies [65,66], affirming the prevalence of significant model uncertainty in such geographically challenging areas.

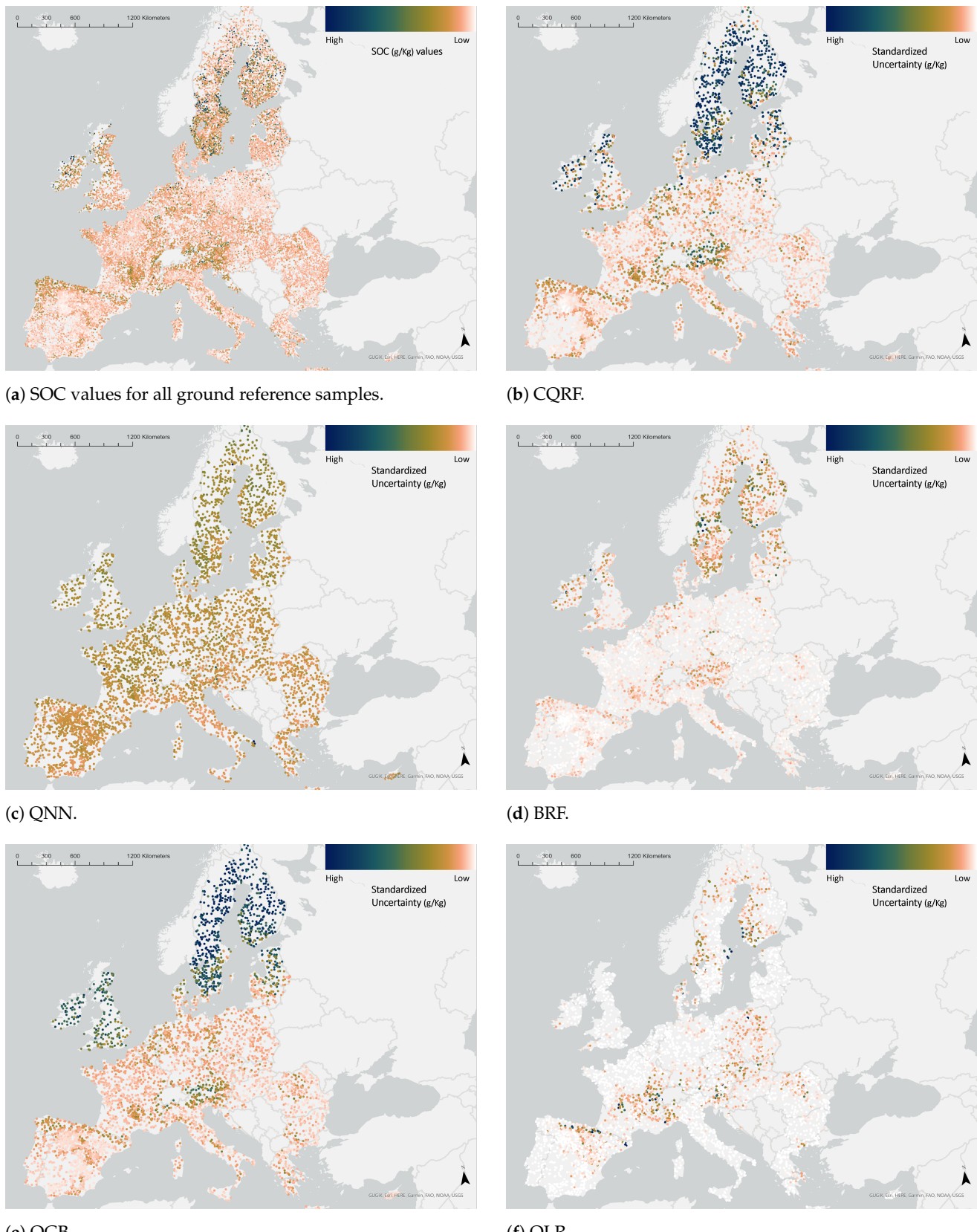

(**a**) SOC values for all ground reference samples.

(**b**) CQRF.

(**c**) QNN.

(**d**) BRF.

(**e**) QGB.

(**f**) QLR.

**Figure 5.** Spatial distribution of SOC values across all ground reference samples (**a**) and the computed standardized uncertainty values from various implemented methods, limited to the test samples of a specific seed (**b**–**f**).

High-altitude regions pose inherent challenges due to their intricate and heterogeneous terrain, making them difficult to access and sample. Consequently, the limited number of soil observations collected in these areas leads to pronounced variability in SOC estimates [67]. This limitation becomes evident when mapping the spatial distribution of soil samples based on their land cover classes, as illustrated in Figure 6. The corresponding histogram of sample counts in Figure 7 underscores the lowest number of samples for the *Wetland* class. Furthermore, it is essential to note that many high-altitude regions in Europe comprise glacier formations, concealing the topsoil and making it challenging to obtain a sufficient number of soil samples.

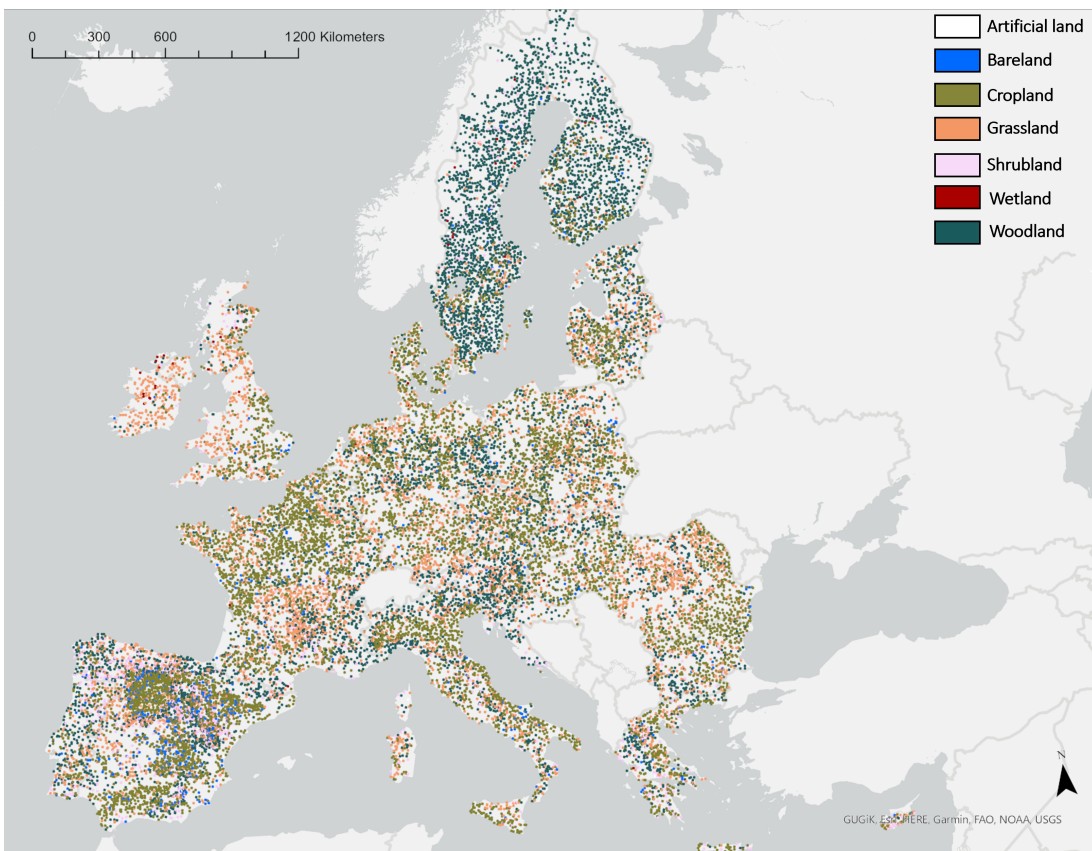

**Figure 6.** Spatial distribution of soil samples based on their land cover classes.

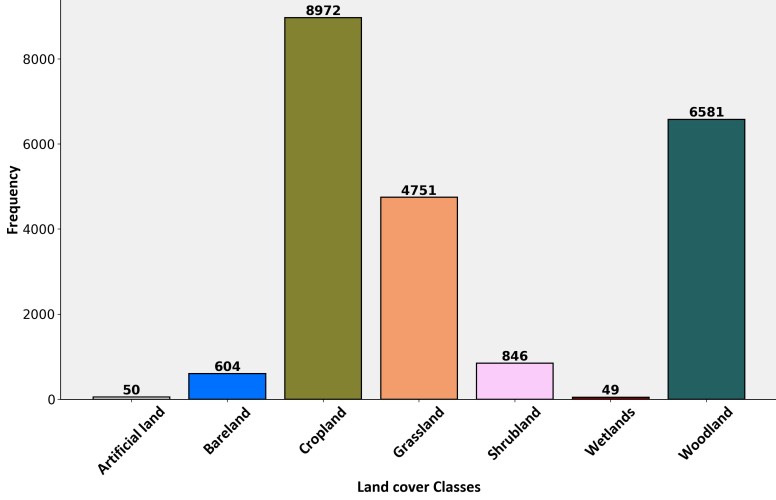

**Figure 7.** Histogram of soil samples for different land cover classes.

Due to decades of cold and anaerobic conditions, peatland soils are very heterogeneous in composition [68]. Such heterogeneity, which is a precursor of varying soil microbial activities and low organic matter decomposition rates [69], makes it difficult to accurately measure SOC and predict its distribution in these regions. Apart from peatlands, soils in boreal regions also have high SOC contents due to decades of accrual and favorable conditions linked to low organic matter decomposition, ample vegetation, and prolonged arctic conditions. These regions not only have ample vegetation but are also highly diverse (e.g., spruce, pines, etc.). Plant diversity constitutes highly heterogeneous soil carbon inputs due to different litter types and undulated topograph [70]. The different litter types tend to decompose at different rates, contributing to SOC variability in these regions. An increase in soil microbial diversity [71] is also an expected positive effect associated with plant diversity, which also influences overall soil composition in boreal regions. Moreover, since some soils in boreal regions are frozen, thawing associated with rapid environmental warming events continues to alter their composition [72]. For instance, as temperature increases, soil microorganisms in boreal regions are becoming more active, which means that greater organic matter decomposition is expected, thus altering SOC contents. All these and other uncovered factors may be contributing to the observed model uncertainties associated with high SOC estimate sampling locations in our study (Figure 5).

Examining the uncertainty patterns derived from QGB (Figure 5e), we observe a striking similarity to the results obtained from the CQRF, aligning with the trends discussed thus far. However, the results from BRF (Figure 5d) share similarities with the previous two models, yet it classifies a substantial number of samples as having very low uncertainty. This characteristic could potentially be misleading, particularly for end-users relying on uncertainty information for decision making. A similar observation holds for the results obtained from QLR (Figure 5f). In the case of QNN (Figure 5c), an intriguing pattern emerges. The model appears to struggle in distinguishing samples with both low and high uncertainties, predominantly categorizing most samples with a mid-level uncertainty. This observation suggests that despite achieving good sample coverage (PICP = 89%), the generated uncertainties might not precisely align with expert knowledge, potentially limiting their utility for end-users.

### 5.2. Empirical Coverage for Low-Sample Classes

As extensively discussed in Section 2, a key aspect of conformal prediction is its statistical guarantee of coverage with an error tolerance $a$. However, the dataset demonstrates imbalances both within the response range (refer to Figure 2) and across various land cover classes, as depicted in Figure 7. The soil samples are distributed across various land cover classes, and the sample sizes exhibit variations among these classes. In particular, wetlands stand out with the lowest sample count (only 49 valid samples for the entire ground truth), accompanied by substantial variability in SOC values within this category (see Figures 6 and 5a). Moreover, several classes, such as artificial land, bare land, and shrubland, have significantly lower sample counts compared to cropland, grassland, and woodland, primarily due to the inherent nature of these three classes. However, from an ML perspective, it becomes challenging for models to accurately capture trends for classes with a limited number of samples. This underscores the importance of the statistical coverage guarantee offered by conformal prediction.

To evaluate the coverage, PICP, provided by each method, especially for classes with a low number of samples, we isolated these samples from the test data. Subsequently, we plotted the response and prediction intervals offered by each implemented method, centering the sample values around 0 for simplicity (see Figure 8). Despite the variability in SOC values, the CQRF demonstrated the ability to cover almost all samples, particularly for the wetlands class, which has only five test samples. This underscores its superiority over QGB. In fact, CQRF emerges as the sole method providing accurate uncertainty estimation that covers wetland samples. When considering other land cover classes with low samples, we observe that the coverage offered by QNN, while high, does not accurately adjust the

true upper and lower bounds of test samples, leading to inaccurate uncertainty values, as discussed in Section 4.5. BRF exhibits a narrow and inaccurate coverage, and as expected, QLR is unable to produce meaningful coverage, confirming the results provided in Table 4.

Although conformal prediction demonstrated superior performance compared to other uncertainty quantification methods, it is not capable of identifying the specific sources of uncertainty—whether originating from the data or the applied models. Further investigation is required to discern the precise origins of uncertainty within the predictive framework.

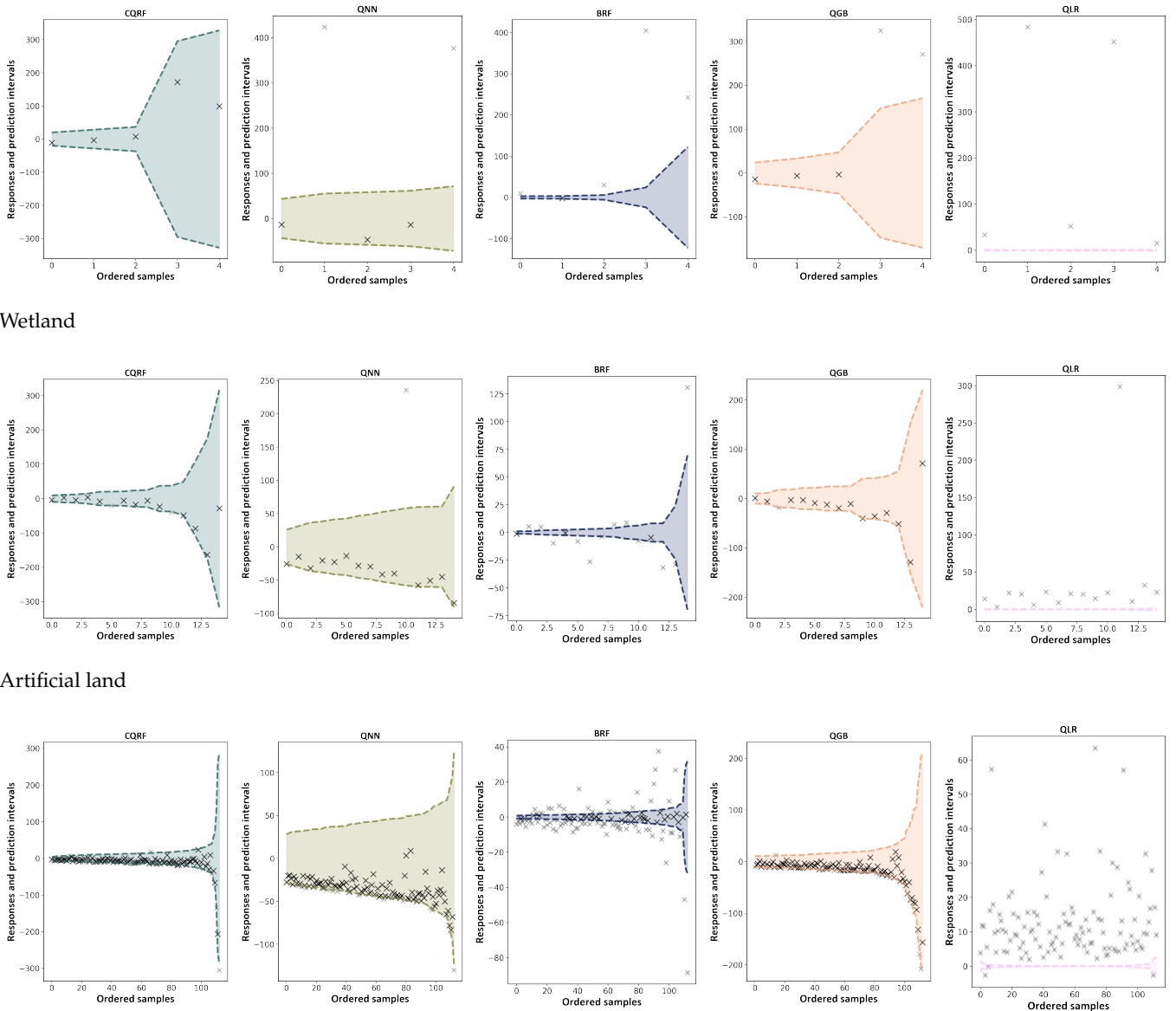

**Figure 8.** *Cont.*

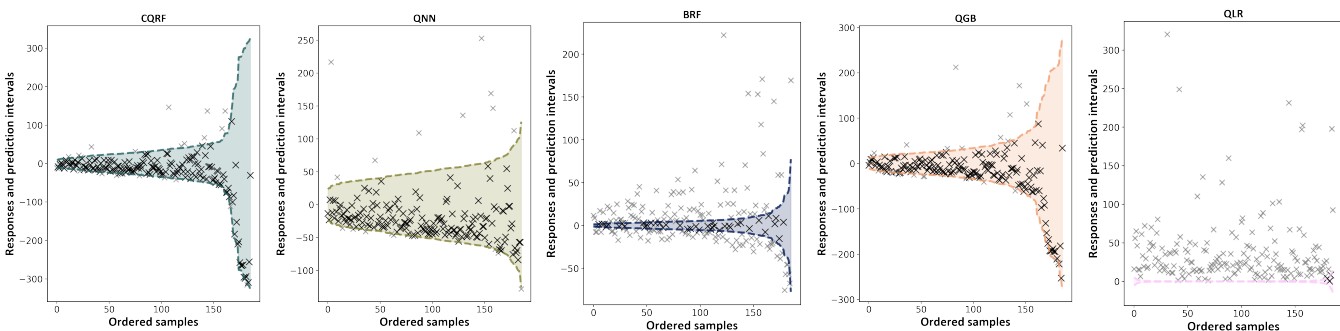

Shrubland

**Figure 8.** The coverage of the test samples for different land cover classes, provided by various implemented methods.

## 6. Conclusions

Conformal prediction, introduced as a method for quantifying uncertainty for SOC estimation in the DSM community, excels in ensuring comprehensive sample coverage. We conducted a comprehensive comparison with four distinct methodologies for uncertainty quantification, leading to the following key conclusions:

- Conformal prediction uniquely demonstrates the ability to effectively adjust prediction intervals derived from an ML regression model. This adaptability ensures the generation of uncertainties that closely align with both empirical observations and expert knowledge derived from the natural processes influencing SOC estimation.
- We empirically demonstrated the coverage efficacy of conformal prediction, even for land cover classes characterized by a limited number of samples. This aspect underscores its versatility and reliability across diverse data scenarios.
- In contrast to inherently time-consuming uncertainty quantification techniques, such as bootstrapping, conformal prediction emerges as an efficient solution. Moreover, its versatility extends beyond being a model-specific approach and can be applied to any ML model.
- Beyond its advantages in uncertainty quantification, conformal prediction demonstrates competitive accuracy metrics, as evidenced by lower RMSE and MAE values compared to other methods. This dual proficiency in uncertainty quantification and accuracy sets it apart from other methodologies.
- The uncertainty maps generated by combining conformal prediction with quantile random forest offer a visually captivating representation of the underlying SOC structure. These patterns align seamlessly with our understanding of SOC formation, providing valuable insights into the intricate dynamics of SOC.

In summary, conformal prediction emerges as a robust and versatile method offering a unique blend of efficient uncertainty quantification, high accuracy, and insightful representations of SOC patterns.

**Author Contributions:** Conceptualization, N.K.; methodology, N.K.; software, N.K.; validation, N.K., N.M.K. and T.S.; formal analysis, N.K. and N.M.K.; investigation, N.K.; resources, T.S.; data curation, N.K.; writing—original draft preparation, N.K., S.A. and N.M.K.; writing—review and editing, N.K., S.A., N.M.K., M.A. and T.S.; visualization, N.K., S.A. and N.M.K.; supervision, T.S.; project administration, N.K.; funding acquisition, T.S. and M.A. All authors have read and agreed to the published version of the manuscript.

**Funding:** We acknowledge the support of the German Research Foundation (DFG) [3150] for the project 'MLTRANS-Transferability of Machine Learning Models in Digital Soil Mapping' (SCHO 739/21-1) and 'Machine Learning for Science' which is part of Germany's Excellence Strategy—EXC number 2064/1—Project number 390727645.

**Data Availability Statement:** The LUCAS-Topsoil samples used during the current study are available upon request from the European Soil Data Center (ESDAC) https://esdac.jrc.ec.europa.eu/resource-type/european-soil-database-soil-properties (accessed on 24 November 2023). The base code https://github.com/moienr/SoilNet (accessed on 17 January 2024) and the the implementation of experiments is available online at https://github.com/nafisehkakhani/Conformal_Prediction_DSM (accessed on 17 January 2024).

**Conflicts of Interest:** Author Meisam Amani was employed by the WSP Environment and Infrastructure Canada Limited. The remaining authors declare that the research was conducted in the absence of any commercial or financial relationships that could be construed as a potential conflict of interest.

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
