# Peer review of "Uncertainty Quantification of Soil Organic Carbon Estimation from Remote Sensing Data with Conformal Prediction"

_remotesensing, doi:10.3390/rs16030438_

Round 1

Reviewer 1 Report

Comments and Suggestions for Authors

This study conducted a comprehensive comparison with four uncertainty quantification methods, and introduced a new conformal prediction of uncertainty method. There is significant novelty in this paper. there are some suggestions as the following.

1. In Table 2, can  you add other information about thd features? such as spatial resolution, time resolution?

2. In Section 5.1, how you set the parameters in the five models? please give them in detail.

3. what is the advantage and difference of the four uncertainty parameters?

Reviewer 2 Report

Comments and Suggestions for Authors

The authors compared different machine learning algorithms and analyzed the uncertainty of remote sensing images in SOC prediction, which has clear research significance. However, the paper is not suitable for publication in the current form, especially for such a high-level journal like Remote Sensing. In particularly, my specific comments and suggestions are listed as follows:

1.The impact of industrialization on carbon in the atmosphere is generally greater than that of soil organic carbon itself. The authors should add an introduction to the impact of soil organic carbon on the ecological environment, or explain the significance of soil organic carbon research from the perspective of human management of soil, carbon sources, and carbon sinks.

2.Since the authors want to evaluate the uncertainty of remote sensing methods in predicting SOC, why is the introduction only introducing the DSM method for measuring SOC, and why is there no review of remote sensing methods for inverting or measuring organic carbon? Yet this part has already been extensively studied, and the prediction results and accuracy are widely applied.

3.I think the structure of this paper is very confusing, and the author should organize the paper according to the general structure of research paper. My opinion is that you can include related works section in the introduction, and merge the sections of Conformal prediction, Data description, preprocessing, and Experiments into a single methods&methodology section.

4.Geographical objects and variables themselves are very complex, which are really influenced by the environment, and thus exhibit significant spatial heterogeneity. However, the basis for choosing European countries as experimental data and their representativeness have not been well explained and analyzed in the manuscript.

5.The introduction of the study area is not specific enough, and the geographical environment, climate conditions, and human factors where the ground reference samples are located are not even explained.

6.The LUCAS program and remote sensing data sources should also be explained in more detail.

7.Section 5.2.2 accuracy assessment, not necessary for this since RMSE and MAE are very common precision evaluation indicators.

8.Why you presented a figure in the conclusion part? Move figure 8 and corresponding text into Discussion section. Only one paragraph is recommended for the conclusion section.

9.More tables are suggested in the paper.

Some other suggestions:

1.Since you have defined many algorithms and models when they first appear in the paper, it is not necessary to provide their abbreviations again in subheadings.

2. Line 255, ‘Digital’ not 'digital'.

Reviewer 3 Report

Comments and Suggestions for Authors

1. The authors evaluated five methodologies for both accuracy and uncertainty and found out the method of conformalized quantile random forest (CQRF) gave statistically reliable results.

2. The content is succinctly described and contextualized with respect to previous and present theoretical background on the topic.

3. The research design, questions, hypotheses and methods are clearly stated.

Round 2

Reviewer 2 Report

Comments and Suggestions for Authors

The reviewer want to congratuate the authors that the manuscript has been improved well. I think my previous comments have been clearly covered and is quite ready for publication in this current form.

Author Response

We express our sincere gratitude for the positive review you provided to us.